# Observational relationships between ammonia, carbon dioxide and water vapor under a wide range of meteorological and turbulent conditions: RITA-2021 campaign

**Ruben B. Schulte[1], Jordi Vilà-Guerau de Arellano[1], Susanna Rutledge-Jonker[2], Shelley van der Graaf[2], Jun Zhang[3], and Margreet C. van Zanten[1,2]**

[1]Meteorology and Air Quality group, Wageningen University & Research, P.O. Box 47, 6700, AA Wageningen, the Netherlands
[2]National Institute for Public Health and the Environment (RIVM), Antonie van Leeuwenhoeklaan 9, 3721, MA Bilthoven, the Netherlands
[3]TNO, Postbus 15, 1755 ZG, Petten, the Netherlands

**Correspondence:** Margreet van Zanten (margreet.van.zanten@rivm.nl)

**Abstract.** We present a comprehensive observational approach, aiming to establish relations between the surface-atmosphere exchange of ammonia ($NH_3$) and the $CO_2$ uptake and transpiration by vegetation. In doing so, we study relationships useful for the improvement and development of $NH_3$ flux representations in models. The $NH_3$ concentration and flux are measured using a novel open-path miniDOAS measurement setup, taken during the five week RITA-2021 campaign (25 August until 12 October 2021) at the Ruisdael Observatory at Cabauw, the Netherlands. After filtering for unobstructed flow, sufficient turbulent mixing and $CO_2$ uptake, we find the diurnal variability of the $NH_3$ flux to be characterized by daytime emissions (0.05 $\mu$g m$^{-2}$s$^{-1}$ on average) and deposition at sunrise and sunset (-0.05 $\mu$g m$^{-2}$s$^{-1}$ on average). We first compare the $NH_3$ flux to the observed gross primary production (GPP), representing $CO_2$ uptake, and latent heat flux ($L_vE$), representing net evaporation. Next we study the observations following the main drivers of the dynamic vegetation response, which are photosynthetically active radiation (PAR), temperature (T) and the water vapor pressure deficit (VPD). Our findings show indication of the dominance of stomatal emission of $NH_3$, with high correlation between the observed emissions and both $L_vE$ (0.70) and PAR (0.72), as well as close similarities in the diurnal variability of the $NH_3$ flux and GPP. However, the efforts to establish relationships are hampered due to the amount of diversity of $NH_3$ sources of the active agricultural region and low data availability after filtering. Our findings show the need to collocate meteorological, carbon and nitrogen studies to advance on our understanding of NH3 surface exchange and its representation.

## 1 Introduction

While nitrogen is an essential nutrient for the growth of plants, acting as a fertilizer, excess nitrogen deposition causes environmental damage and leads to an increased public health risk through the formation of particulate matter (Bobbink et al., 2003; Behera et al., 2013; Erisman and Schaap, 2004; Erisman et al., 2013; Smit and Heederik, 2017). When nitrogen critical loads are exceeded, excess nitrogen deposition threatens biodiversity through acidification and eutrophication of soils. When mitigation of the harmful effects of nitrogen fails, there can be serious political, economic and societal consequences, as demonstrated by the current Dutch nitrogen crisis (Stokstad, 2019). Atmospheric ammonia ($NH_3$) plays a key role in the deposition of nitrogen, mainly originating from agricultural activity. This is especially true in the Netherlands where $NH_3$ deposition accounts for about three-quarters of all nitrogen deposition (Wichink Kruit and van Pul, 2018; RIVM et al., 2019).

Efforts to mitigate the harmful effects of nitrogen deposition heavily rely on models representing the concentration and deposition of nitrogen compounds, supported by a network of concentration and surface-atmosphere exchange measurements. The surface-atmosphere exchange in such models is represented by parameterizations, which are developed, validated and improved based on advanced high-resolution observations. In the case of atmospheric ammonia, taking accurate high-resolution measurements is notoriously difficult, due to the reactive nature of gaseous $NH_3$ causing the gas to "stick" to inlet walls of conventional instruments (Parrish and Fehsenfeld, 2000; von Bobrutzki et al., 2010). These challenges are amplified when measuring the $NH_3$ surface-atmosphere exchange flux (deposition or emission), where high precision is particularly important (Nemitz et al., 2004; Whitehead et al., 2008).

Recent developments in advanced instrumental techniques resolve these inlet issues by using optical open-path analyzers. Swart et al. (2023) presents an intercomparison of two novel open-path measurement setups, aimed at measuring the $NH_3$ flux at half-hourly resolution: the RIVM-miniDOAS 2.2D and the commercial Healthy Photon HT8700E. The two setups showed very similar results, despite being widely different in their measurement principle and approach to derive the flux from concentrations, as the Healthy Photon uses the eddy covariance technique while the miniDOAS applies the flux-gradient method to line average concentration measurements over a 22 m open-path at two heights. In this study, we continue the analysis of the observations of the miniDOAS system presented by (Swart et al., 2023), as the system provides reliable measurements of both the concentration and flux with a high operational uptime.

In a previous study, based on measurements from the predecessor of the miniDOAS system at the Veenkampen meteorological site in the Netherlands, we identified that the mechanisms behind stomatal exchange of $NH_3$ are not yet fully understood (Schulte et al., 2021). Here, we continue to study this stomatal exchange pathway by linking the observed $NH_3$ flux ($F_{NH_3}$) to photosynthesis, i.e. the stomatal exchange of $CO_2$ and water vapor (plant transpiration). The similarities between the stomatal exchange of $NH_3$ and $CO_2$ have long been recognized (San José et al., 1991; Schrader et al., 2020). However, there are very few parallel measurements of $NH_3$ and $CO_2$ fluxes, and research into the two gases is generally conducted by separate scientific communities (Milford et al., 2001). Milford et al. (2001) performed one of the few attempts to develop a simple parameterization for both the $CO_2$ and $NH_3$ flux, but was unsuccessful to find such relations for $NH_3$ as the observed $NH_3$ flux over Scottish Heathland was dominated by non-stomatal exchange. Further, Zöll et al. (2019) performed an analysis to study whether the biosphere-atmosphere exchange of total reactive nitrogen was driven by the same variables as carbon dioxide.

Our aim is to relate $NH_3$ and $CO_2$ fluxes to advance in our understanding of $NH_3$ stomatal exchange. These surface exchanges require to be related to the sensible and latent heat fluxes, and the diurnal boundary layer dynamics (Vilà-Guerau de Arellano et al., 2023). Utilizing the recent developments in $NH_3$ measurement techniques, we combine the high-quality miniDOAS $F_{NH_3}$ observations with measurements of both $CO_2$ and water vapor fluxes, as well as other meteorological variables. As our data set is limited due to the diversity of weather conditions and the complexity associated to nearby multiple sources of ammonia, our analysis acts as a proof of concept. Serving as an example for the need of combined high quality $NH_3$ flux measurements with auxiliary measurements of $CO_2$, water vapor fluxes and other meteorological variables. As such we decided to guide our analysis solely using observations and keep the use of representation of processes to interpret our data to a minimum. We first describe the observations, after which we link the observed $F_{NH_3}$ to stomatal exchange, with the intention to establish relationships between the stomatal exchange of ammonia and the processes of $CO_2$ uptake and transpiration by vegetation. As these processes of photosynthesis are well understood, we explore how this understanding can lead to further improving the parameterization of the $NH_3$ stomatal exchange.

## 2   Characterizing the RITA-2021 campaign observations

### 2.1   Site description and measurement strategy

In September 2021, the Ruisdael Land-Atmosphere Interactions Intensive Trace-gas and Aerosol measurement campaign, known as RITA-2021, took place at the Cabauw Observatory (https://ruisdael-observatory.nl/cabauw/). The Cabauw Observatory, one of the 6 sites within the Ruisdael Observatory, is located on flat grassland in the Netherlands ($51.971^oN$, $4.927^oE$), with an average grass height of 0.1 m. The site provides a unique set of surface and upper air observations, matched by only a very few station world-wide. This includes measurements of thermodynamic variables along the 213 m mast, radiation, surface fluxes, clouds and trace gasses. Surface elevation changes are at most a few meters over 20 km and the nearby region is agricultural. An overview of the Cabauw site, the instruments stationed at the site and its 50-years of observations is given in Bosveld et al. (2020).

During the campaign, 48 days of ammonia measurements are taken using the miniDOAS (Differential Optical Absorption Spectroscopy) flux measurement setup (Berkhout et al., 2017), starting on 25 August until 12 October. The measurement setup and more details on the measurement campaign are described in Swart et al. (2023). In short, the miniDOAS is an

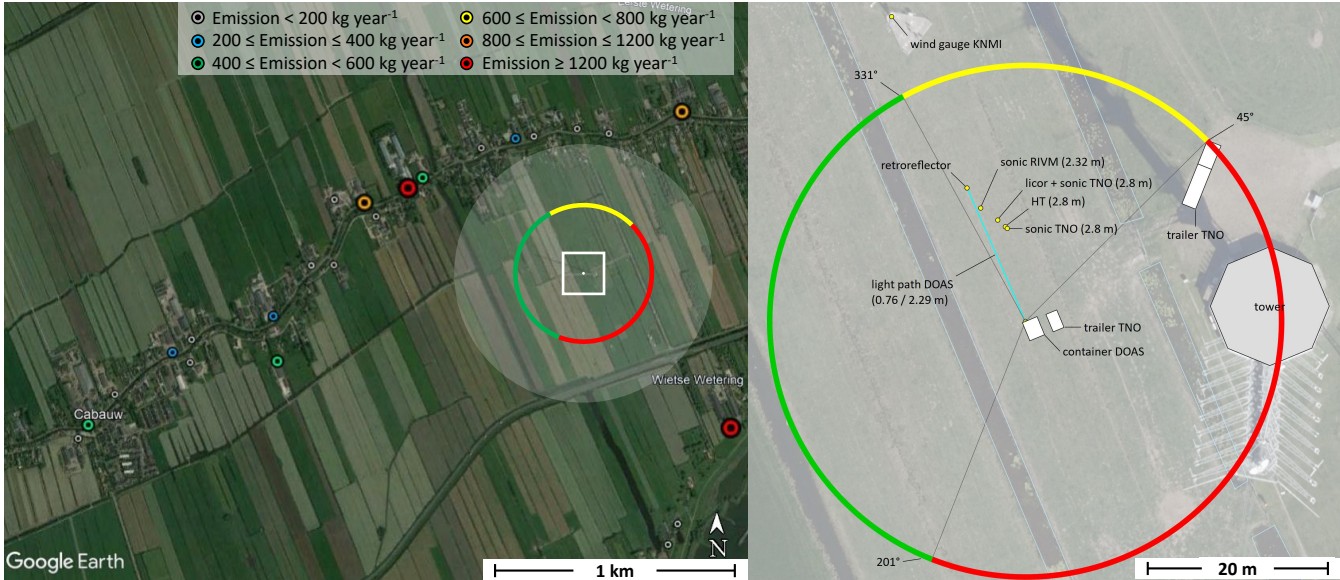

**Figure 1.** The area surrounding the Cabauw Observatory (left) and the setup of the instruments at the measurement site (right). The transparent white circle represents a distance of 500 m from the NH$_3$ measurements and the color coded dots represent the locations of nearby farms whereby the emission strength is specified in kg NH$_3$ per year (Source: Emissieregistratie, www.emissieregistratie.nl, last access 21 January 2022). The colored circle in both panels indicates the wind directions where the airflow towards the miniDOAS light-path is obstructed by either other instruments (yellow) or larger structures such as the tower and containers (red). (Source left panel: "Cabauw Observatory", 51.971$^o$N, 4.927$^o$E. © Google Earth, 27 January 2022, Image by Landsat / Copernicus. Source right panel: Swart et al. (2023), modified)

optical instrument, measuring the line average concentration (mass density) over a 22 m open path from instrument to its retroreflector. The 30 minute average NH$_3$ concentrations have an accuracy of 3% (e.g. 0.15 $\mu$g m$^{-3}$ at the median NH$_3$ concentration of 5 $\mu$g m$^{-3}$ during the campaign, for further details see Swart et al. (2023). The flux measurement setup uses two miniDOAS instruments, which measure the concentration over parallel paths at different heights, i.e. 0.76 m and 2.29 m respectively. Regular intercalibrations between the miniDOAS instruments allowed quantification of and correction for any potential bias between the two instruments. The remaining random uncertainty in the $\Delta$ NH3 was 0.088 $\mu$g m$^{-3}$ (1$\sigma$; for further details see Swart et al. (2023). F$_{NH_3}$ is then inferred using the flux-gradient method, based on the Monin–Obukhov similarity theory (Moene and Van Dam, 2014). The flux-gradient method combines the observed vertical NH$_3$ gradient with turbulent measurements of a sonic anemometer (model Gill WindMasterPro$^{TM}$ Gill Instruments, Lymington, UK) (Wyers et al., 1993; Nemitz et al., 2004; Wichink Kruit et al., 2007; Schulte et al., 2021). The sonic anemometer was mounted at 2.8 m above the ground alongside the mini-DOAS measurement path. Temperature data is based on the corrected air temperature as calculated by the EddyPro software from the sonic data. The 10 Hz open-path H$_2$O and CO$_2$ analyser (LI-7500DS, LI-COR Biosciences,Lincoln, USA) was placed at a similar height, 15 cm away from the sonic (for more details see information over sonic #1 in (Swart et al., 2023)) The CO$_2$ and water vapor fluxes and other micrometeorological parameters were calculated using EddyPro software (LI-COR Bio-sciences) at 30-min intervals using the 10 Hz raw data. The flux calculation procedure followed the general best practices as applied across the FluxNet network (e.g. Mauder et al. (2022)) including co-ordinate rotation (Wilczak et al., 2001), spectral corrections for both filtering (Moncrieff et al., 2004) and low pass filtering (Moncrieff et al., 1997) and addition of the Webb–Pearman–Leuning density term (Webb et al., 1980).

The measurement field and its surroundings are shown in Fig. 1. The miniDOAS light paths are aimed in north-northwestern direction (right panel) to ensure unobstructed flow for wind coming from the west, which is the dominant wind direction in the Netherlands. North of the light path, shown in yellow in Fig 1, the flow of air is obstructed by several instruments, including the aforementioned sonic anemometer. To the east and south, the airflow is obstructed by a trailer, the 213 m high meteorological tower and the container which houses the miniDOAS instruments. The unobstructed region west of the measurement field, is mainly characterized by actively managed agricultural grassland and the small town of Cabauw (about 750 inhabitants), as shown left in Fig. 1. Several farms can be seen northwest and west of the measurement field, with varying emission strengths up to over 1200 kg NH$_3$ year$^{-1}$. Sheep and cattle graze on these agricultural fields, which are actively maintained and fertilized.

**Table 1.** Filter criteria, being applied in sequence; with filter acceptance rates (in percentages and hours).

| # | Filter | Criterion | Acceptance [%] | [hours] |
|---|--------|-----------|:-------------:|:-------:|
| - | Unfiltered observations | - | 100 % | 1152 |
| Discard 1: | miniDOAS intercalibration | - | 65 % | 746 |
| Discard 2: | Fertilization events 11 - 12 September | - | 61 % | 698 |
| Filter 1: | Wind direction | $331^o \geq U_d \geq 201^o$ | 16 % | 188 |
| Filter 2: | Rain duration | $t_{rain} \leq 5\,min$ | 16 % | 179.5 |
| Filter 3: | Turbulent mixing | $u_* > 0.1\,ms^{-1}$ | 13 % | 151 |
| Filter 4: | Gross primary production | $GPP > 0\,mgCm^{-2}s^{-1}$ | 11 % | 123.5 |
| Filter 5: | Incoming short-wave radiation | $SW_{in} > 10\,Wm^{-2}$ | 9 % | 102 |

These activities were not documented and sporadic fertilization events do affect the NH$_3$ measurements, as will be discussed later.

## 2.2 Data filtering

We apply several filter criteria to the RITA-2021 observations, which are shown in Table 1 with acceptance rates for each individual filter criterion. The miniDOAS flux setup requires several days of intercalibration measurements, as described in Swart et al. (2023). No ammonia flux can be inferred from these intercalibration measurements, leaving 65% of the campaign observations suitable for flux measurements. We furthermore discard observations from 11 - 12 September, as these NH$_3$ emission fluxes are outliers with respect to the average observed NH$_3$ flux, indicating towards a fertilization event in close proximity to the measurement site.

The remaining measurements are processed by applying 5 filters in total. The use of the flux-gradient method requires unobstructed upwind air flow with sufficient turbulent mixing. Figure 1 shows that the instruments were positioned anticipating winds from the south-west (green), with the obstacles located east (red) and north (yellow) of the miniDOAS optical path. We therefore apply a criterion filtering for wind directions between 201º and 331º. This filter leads to a large reduction of data available for analysis, decreasing the available data from 61 % to 16 % as the prevalent wind direction during the campaign was from the north-east. As a secondary effect of this filter, the available observations are taken under synoptic weather conditions characterized by frontal passages with some rain events. The second filter excludes rain events lasting more than 5 min, as rain droplets can obstruct the light path of the miniDOAS. Finally, sufficient turbulent mixing is one of the main requirements for flux measurement using the flux-gradient method. The third filter therefore requires the friction velocity to have a value of at least 0.1 m s$^{-1}$ (u$_* \geq$ 0.1 m s$^{-1}$). With these three filters, we ensure the quality of the ammonia measurements, observing the NH$_3$ flux with an average precision of 0.015 $\mu$g m$^{-2}$s$^{-1}$ (1$\sigma$; for further details see (Swart et al., 2023).

The fourth and fifth filter criteria focus on the ammonia surface-atmosphere exchange pathways. The NH$_3$ flux follows three pathways: the stomatal pathway, external leaf surface pathway and the soil pathway (Nemitz et al., 2001; Massad et al., 2010; van Zanten et al., 2010). The latter is generally assumed to be negligible for the F$_{NH_3}$ over grass, as the dense vegetation completely covers the soil. The external leaf pathway, represents exchange of ammonia with a thin film of water and leaf surface waxes on the leaf surface, and depends on the relative humidity (RH) (Van Hove et al., 1989). Finally, the stomatal pathway represents exchange of NH$_3$ through the plant stomata with ammonium dissolved in the apoplast fluids of the plant (Farquhar et al., 1980; Wichink Kruit et al., 2010). These processes occur at the leaf scale (micrometer or millimeter level) and as such require a representation of photosynthesis and stomatal aperture that requires to be evaluated with observations (Vilà-Guerau de Arellano et al., 2020). The upscaling to the canopy level, allows it to be compared with observations inferred from eddy-covariance such as GPP (Filter 4).

The NH$_3$ exchange through the stomatal pathway is governed by the dynamic response of vegetation to meteorological conditions and is closely related to photosynthesis. The stomata open during the day in response to solar radiation, as the vegetation uses energy for photosynthesis, particularly the photosynthetically active radiation (PAR) (Hsiao, 1973; Cowan and Farquhar, 1977; Papaioannou et al., 1996; Ronda et al., 2001). Plants ingest CO$_2$ through the stomata, but water from inside the plant can evaporate as the stomata are opened. The plant can reduce this loss of water by (partly) closing the stomata in case of high water vapor pressure deficit (VPD), or increase the evaporation rate by actively opening the stomata. Increasing the evaporation rate provides cooling, lowering the leaf temperature to reach optimal conditions to perform photosynthesis (Jacobs and de Bruin, 1997; Takagi et al., 1998; de Groot et al., 2019; Vilà-Guerau de Arellano et al., 2020). As the temperature and VPD are often highest in the afternoon, the stomata often partly close to manage the loss of water. During the night, there is no PAR for photosynthesis, so the stomata are closed. As a result, the characteristics of ammonia surface-atmosphere exchange

**Table 2.** A characterization of the meteorology of the 17 unique days at which observational data passes the filters, with the 17 day average and the range of the diurnal minimum/maximum of several (meteorological) variables. Daily maximum flux footprint length (70%) refers to the maximum upwind distance in meters encompassing the source area that contributed 70% of the measured flux. For GPP and flux footprint length night time are excluded.

| Variable | Symbol | Diurnal minimum/maximum | | | |
|---|---|---|---|---|---|
| | | 17-day average | | 17-day range | |
| Daily minimum temperature | $T_{min}$ | 11.5 | °C | 5.6 - 16.7 | °C |
| Daily maximum temperature | $T_{max}$ | 19.7 | °C | 13.6 - 25.5 | °C |
| Daily maximum wind speed | u | 4.5 | m s$^{-1}$ | 2.2 - 7.2 | m s$^{-1}$ |
| Daily maximum net radiation | $Q_{net}$ | 295 | W m$^{-2}$ | 137 - 400 | W m$^{-2}$ |
| Daily maximum sensible heat flux | H | 99 | W m$^{-2}$ | 27 - 173 | W m$^{-2}$ |
| Daily maximum latent heat flux | $L_vE$ | 145 | W m$^{-2}$ | 83 - 230 | W m$^{-2}$ |
| Daytime maximum gross primary production | GPP | 0.78 | mgC m$^{-2}$ s$^{-1}$ | 0.57 - 1.4 | mgC m$^{-2}$ s$^{-1}$ |
| Daily maximum water vapor pressure deficit | VPD | 966 | Pa | 365 - 1420 | Pa |
| Daytime maximum flux footprint length (70 %) | $fp_{70\%}$ | 148 | m | 88 - 255 | m |

differ between day and night, with the stomatal pathway being dominant during the day and the external leaf pathway being the dominant pathway during the night and in the early morning.

The uptake of $CO_2$ is represented by the Gross Primary Production (GPP), in mgC m$^{-2}$s$^{-1}$. The GPP and the ecosystem respiration (ER) combined define the Net Ecosystem Exchange (NEE) of $CO_2$. Using the sign convention that the flux towards the surface is positive, we define the net ecosystem exchanges as: $NEE = GPP + ER$ where under normal daytime grassfield conditions our observations are NEE > 0, and the inferred GPP are positive and ER negative, respectively. The ecosystem respiration is estimated by taking the average campaign nighttime (defined when the net available radiation is zero, $Q_{net} < 0$) $CO_2$ flux, which is approximately - 0.6 mgC m$^{-2}$ s$^{-1}$. The GPP is then estimated by combining the observed $CO_2$ flux with the estimated respiration.

The approach described above, fits with our aim to guide the analysis by measurements only. However, well-established methods exist to partition NEE into GPP and ER. In appendix A we show that using the Arrhenius-type relationship between temperature and nighttime $CO_2$ flux to describe ER as proposed by Lloyd and Taylor (1994), and then subtracting that from NEE to arrive at GPP, only changes the GPP estimates slightly. Because of its limited impact on the results, we continue with the observation-based estimate of GPP in the main text.

To capture observations with active stomatal exchange, Filter 4 is set to only accept GPP > 0 mgCO$_2$ m$^{-2}$ s$^{-1}$. Due to the uncertainty of our GPP estimate, there are still some night-time observations which pass the filter. We therefore add an additional 5[th] filter using incoming shortwave radiation (SW$_{in}$). Only measurements with SW$_{in}$ > 10 W m$^{-2}$ will pass, in order to filter out these last remaining night-time observations.

After filtering, 102 hours (9 %) of all RITA-2021 observations, or 18 % of all daytime RITA-2021 observations, are available for analysis. These observations are taken over 17 unique days, spanning 29 August to 30 September, with an average of 6 hours and a maximum of 12 hours of accepted measurement per day.

## 2.3 Characterization of the campaign meteorology

The summer months (June, July and August) leading up to the RITA-2021 campaign are characterized as an average Dutch summer, with average temperatures (17.7 °C), above average precipitation (244 mm accumulated) and below average hours of sunshine (618 hours). Additionally the ground and surface water levels are actively managed in order to sustain optimal conditions for the agricultural activity in the area (Brauer et al., 2014). It is therefore expected that the role of long-term vegetation stress on stomatal exchange is negligible during the RITA-2021 campaign.

As discussed in Section 2.2, high temperatures or VPD can induce vegetation stress during the campaign. We therefore characterize the meteorological conditions of the 17 unique days in which the 102 hours of filtered measurements were taken. The meteorological conditions of these days are summarized in Table 2, which shows the 17 day average and the observed range of the diurnal minimum/maximum of several variables. The 17-day average values provide a characterization of mild meteorological conditions with no indication that the vegetation is under stress. Additionally, Table 2 includes an estimate of the maximum daytime footprint determined using the sonic anemometer fluxes at a height of 2.8 m, following the method from Kljun et al. (2015). This footprint refers to the maximum upwind distance in meters encompassing the source area that contributed 70% of the measured flux and serves as a first-order approximation of the footprint of the NH₃ flux measurements.

As the filtered campaign measurements are characterized by frontal passages, the weather conditions range from clear sky summer conditions with moderately high temperatures, to colder cloudy days with short precipitation events (not shown). Furthermore, the atmospheric stability for the 102 hours of filtered measurements is classified using the measured Obukhov length (L) and the height of the sonic anemometer (z = 2.8 m). In total, 4.5 hours (4 %) can be classified as stable (z/L > 0.05),
61 hours (60 %) as neutral (-0.05 $\leq$ z/L $\leq$ 0.05) and 36.5 hours (36 %) as unstable conditions (z/L < -0.05). This variation leads to a large spread in all variables shown in Table 2, as indicated by the column showing the 17-day range.

## 2.4   General characterization of the $NH_3$ observations

The variety in meteorological conditions could be an explanation of the large day-to-day difference in the observed $NH_3$ concentrations, shown in Fig. 2b. The histogram is highly skewed and shows that most observed $NH_3$ concentrations are below
10 7 $\mu$g m$^{-3}$, however higher concentrations with a maximum value of 24.7 $\mu$g m$^{-3}$ are also present. Still, the mean (solid line) and median (dotted line) concentrations do indicate that the concentration decreases during the day, until the late afternoon. This would be in line with observations at several other sites, both in the Netherlands (Wichink Kruit et al., 2007; Schulte et al., 2021) and in other countries, e.g. Scotland (von Bobrutzki et al., 2010) or Italy (Ferrara et al., 2021). The large day-to-day differences in the $NH_3$ measurements could be a result of the changing meteorological conditions, the nearby agricultural
activity, or a combination of both.

Despite the high variability in the $NH_3$ concentration measurements, a consistent diurnal variability is observed in the $NH_3$ gradient ($\Delta NH_3$) and corresponding flux in Fig. 2c and 2d. Both Fig. 2b and 2c indicate that the observed $\Delta NH_3$ is independent of the absolute $NH_3$ concentration, i.e. high absolute concentrations do not lead to a large concentration difference between the two miniDOAS instruments. The average diurnal variability is characterized by negative $\Delta NH_3$ (deposition) in the early
morning and late afternoon and positive (emission) $\Delta NH_3$ during the afternoon, with a typical range of about 0.5 $\mu$g m$^{-3}$ in both directions. In total 79 % of the filtered observations have a positive $\Delta NH_3$, corresponding to $NH_3$ emissions.

As $F_{NH_3}$ is directly inferred from $\Delta NH_3$, the diurnal variability in Fig. 2c and d is very similar. The $NH_3$ flux typically reaches its maximum around noon at little over 0.05 $\mu$g m$^{-2}$s$^{-1}$ on average, with individual noon observations ranging from -0.01 $\mu$g m$^{-2}$s$^{-1}$ to 0.14 $\mu$g m$^{-2}$s$^{-1}$. Note that the measurements taken on 11 - 12 September, the aforementioned fertilization
event, are approximately a factor 4 larger than the mean campaign values. Despite the large observed $F_{NH_3}$ on these days, the observed concentrations are only slightly larger than the campaign averages. These two days will not be included in the analysis presented in this study, but they are shown as an illustration of how fertilization events can impact our analysis.

## 2.5   Characterization of the ammonia flux

In Fig. 3a, we show the observed ammonia flux against the air temperature, with the colors indicating the atmospheric $NH_3$
concentration at 2.29 m. Despite our efforts to filter for observations where the stomatal pathway is dominant, it cannot be ruled out that the external leaf pathway still plays an important role in the morning, through deposition onto morning dew at canopy level (van Zanten et al., 2010; Wentworth et al., 2016). We therefore use black circles to mark observations taken before 12:00 UTC with RH > 80 % in Fig. 3. These highlighted observations indeed generally correspond with measurements of deposition or weak emission, indicating that $NH_3$ exchange through the external leaf pathway is still significant for these observations.
While the inclusion complicate our analysis of stomatal $NH_3$ exchange, they are still included in the analysis as it also offers an opportunity to test if the relations found in the filtered dataset differ for the marked- and unmarked observations. If that is the case, it shows we are indeed able to attribute the unmarked observations to stomatal exchange.

Figure 3a shows $F_{NH_3}$ to increases with temperature for low atmospheric concentration (2 $\mu$g m$^{-3}$ $\leq$ $NH_{3, 2.29m}$ $\leq$ 7 $\mu$g m$^{-3}$). We attribute this increase in $NH_3$ emissions to the change in $\Delta NH_3$ for increasing temperature, i.e. the difference between
40 the approximately constant atmospheric $NH_3$ concentration and stomatal compensation point. Following parameterizations of this compensation point, we find it related to the (leaf) temperature and some form of nitrogen availability parameter (e.g. actual or long-term $NH_3$ concentration); increasing non-linearly with increasing temperature or nitrogen input (Nemitz et al., 2001; Massad et al., 2010; van Zanten et al., 2010). In Fig. 3b a theoretical stomatal compensation point (dotted line) is added. Calculated following the DEPosition of Acidifying Compounds (DEPAC) parameterization (van Zanten et al., 2010),
using air temperature and campaign median $NH_{3, 2.29 m}$ (7.7 $\mu$g m$^{-3}$). $F_{NH_3}$ shows more scatter for measurements taken at high temperatures (> 21 °C). While Fig. 3b shows only small variations in the $NH_3$ concentration for temperatures below 21 °C, $NH_3$ concentration for these warmer temperatures are higher than the campaign average (> 7 $\mu$g m$^{-3}$) and highly variable. As the $NH_3$ flux is directly related to the difference between the atmospheric $NH_3$ and the stomatal compensation point, the variability in the atmospheric concentration lead to the scatter shown in Fig. 3a, where higher $NH_3$ concentrations correspond
to weaker emission fluxes.

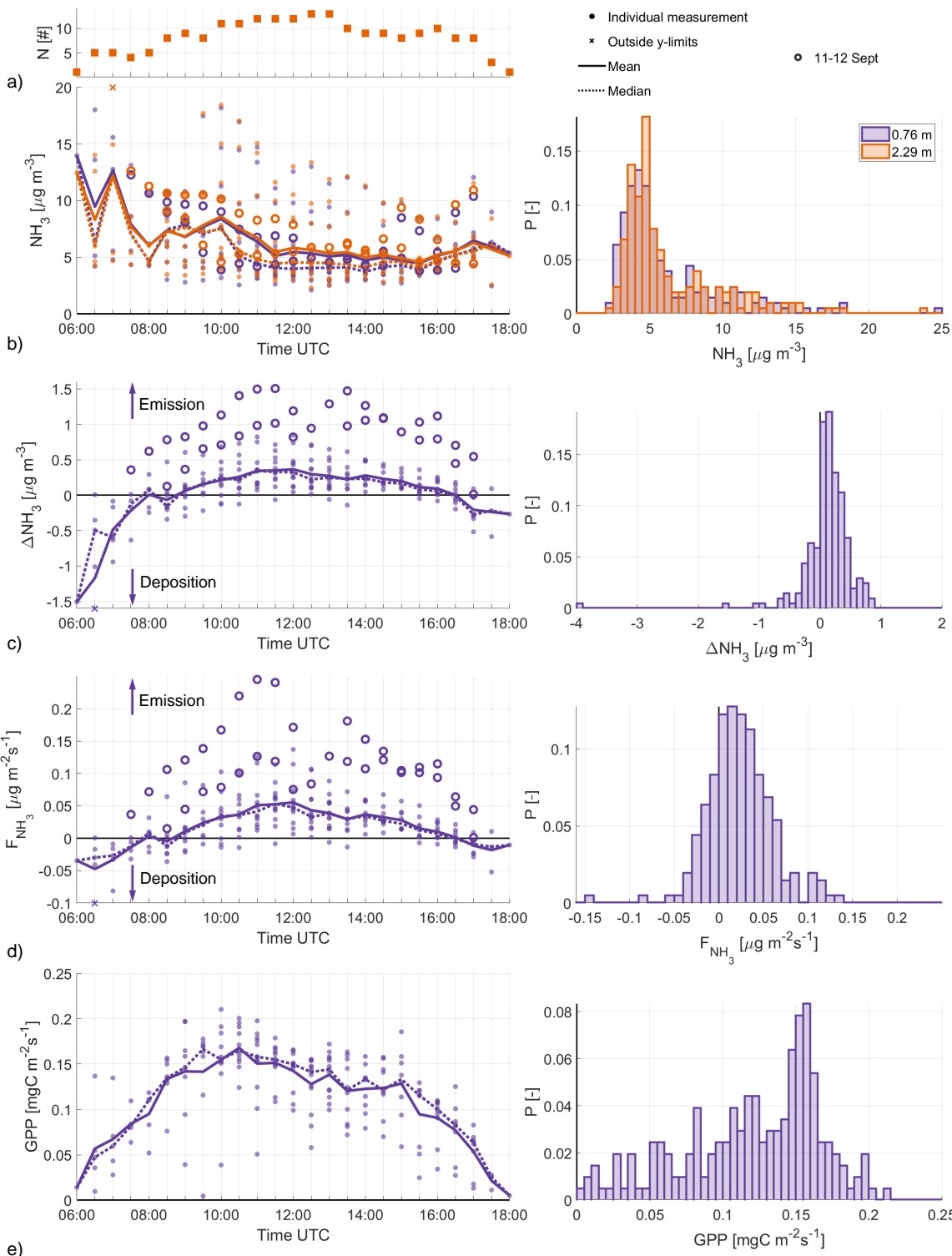

**Figure 2.** The diurnal variability from sunrise (6:00 UTC) to sunset (18:00 UTC) of the filtered $NH_3$ concentration (b), $NH_3$ gradient (c), $F_{NH_3}$ (d) and the GPP with the corresponding histogram to the right. At each moment in time, the multi-day mean (solid line) and median (dotted line) are calculated. Highlighted are observations from the fertilization event at 11 - 12 September (open circles). The N number of observations over which these averages are calculated are displayed at the top (a). $\Delta NH_3$ is defined so the sign matches that of $F_{NH_3}$ i.e., negative numbers indicate deposition and positive numbers indicate emission.

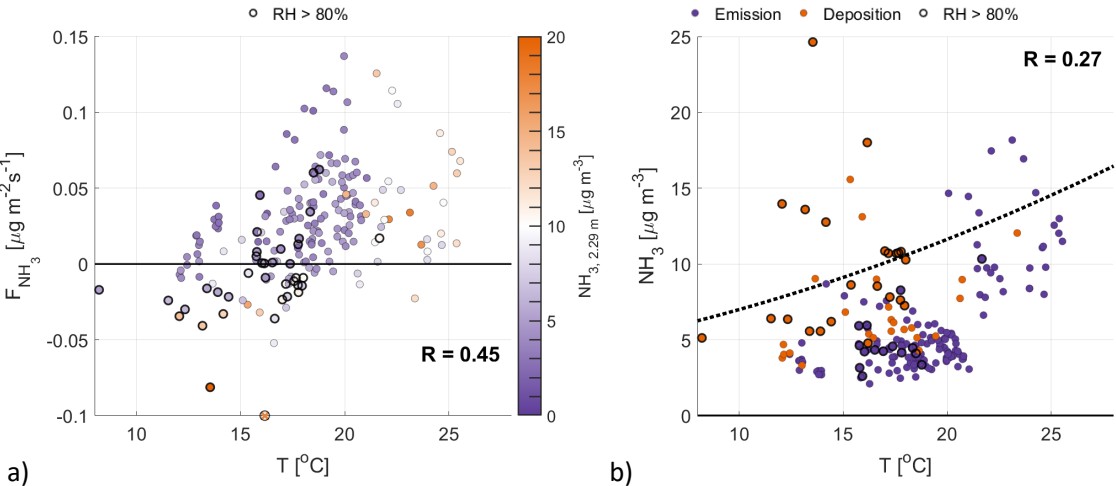

**Figure 3.** The 2.8 m temperature plotted against $F_{NH_3}$ (a) and observed $NH_{3, 2.29 m}$ concentration (b). The color coding in (a) represents the $NH_3$ concentration, observed at 2.29 m. In (b), the dotted line represents the theoretical stomatal compensation point ($\chi_s$) for a long-term $NH_3$ concentration of 7.7 $\mu g$ m$^{-3}$. Highlighted with black circles are observations with RH > 80%, taken before noon, where $NH_3$ exchange through the external leaf pathway can play a significant role.

## 3    Ammonia flux relationships to dynamic vegetation responses

The diurnal pattern of $F_{NH_3}$ in Fig. 2d shows similarities with the diurnal variability of the GPP in Fig. 2e. To further study the role of stomatal exchange during the campaign, we link the observed $F_{NH_3}$ to the dynamic vegetation responses. First, we relate the ammonia flux to the GPP, the latent heat flux ($L_vE$) and the sensible heat flux (H). The GPP and (the transpirational
part of) $L_vE$ are directly governed by the opening and closing of the stomata and represent stomatal exchange. Given the low data availability (9%), we are aware that the analysis could be dominated by variations resulting from the diurnal variability of the fluxes. We therefore also include H in our analysis. The sensible heat flux is only indirectly related to the dynamic vegetation response through the surface energy balance, as the available energy from (solar) radiation and the soil heat flux is split between $L_vE$ and H. If the observed fluxes are indeed regulated through the opening and closing of stomata, the analysis
for $F_{NH_3}$ to $L_vE$ and GPP should differ from the comparison with H.

   Next, we organize the observations following current dynamic vegetation models, based on temperature, radiation and moisture (Jarvis et al., 1976; Stewart, 1988; Ronda et al., 2001). Here, we compare the responses of the four individual fluxes to temperature (T), PAR and VPD. As in the models these three variables control the stomatal response at canopy level, we will use the responses of the fluxes to these variables as a guidance to better understand the diurnal variability of the ammonia flux.
Note that measurements taken on 11 - 12 September are not used to calculate correlation coefficients, but are shown in the figures and included in the visual analysis.

### 3.1    Relating the ammonia flux to photosynthesis

Plotting $F_{NH_3}$ against the GPP in Fig. 4a shows a low positive correlation between the two fluxes, with a correlation coefficient of 0.48. There is a large spread in the data, particularly for GPP values larger than 0.125 mgC m$^{-2}$s$^{-1}$. Part of this spread is
attributed to the high relative humidity (black circles), where $F_{NH_3}$ is not yet dominated by stomatal exchange and the external leaf pathway is still expected to be significant. Note that the atmospheric stability (color coded) plays an important role for the GPP as unstable conditions are typically characterized by clear skies and high PAR values, which favors photosynthesis (discussed in Section 3.2). This relation is not found in the observed $F_{NH_3}$, as there is a large spread in $F_{NH_3}$ for both neutral and unstable conditions.
A moderate positive correlation is found in Fig. 4b between $F_{NH_3}$ and $L_vE$. Our interpretation of this moderate correlation is that both transpiration and stomatal $NH_3$ emissions follow a similar process. The opening of the stomata for photosynthesis allows for the exchange of several gasses, including water vapor and ammonia, depending on VPD or the difference between atmosphere $NH_3$ and the stomatal compensation point (Cowan and Farquhar, 1977; Hsiao, 1973; Farquhar et al., 1980; Wichink Kruit et al., 2010). Note that $L_vE$ represents the net evaporation (Miralles et al., 2020) since evaporation from the soil plays

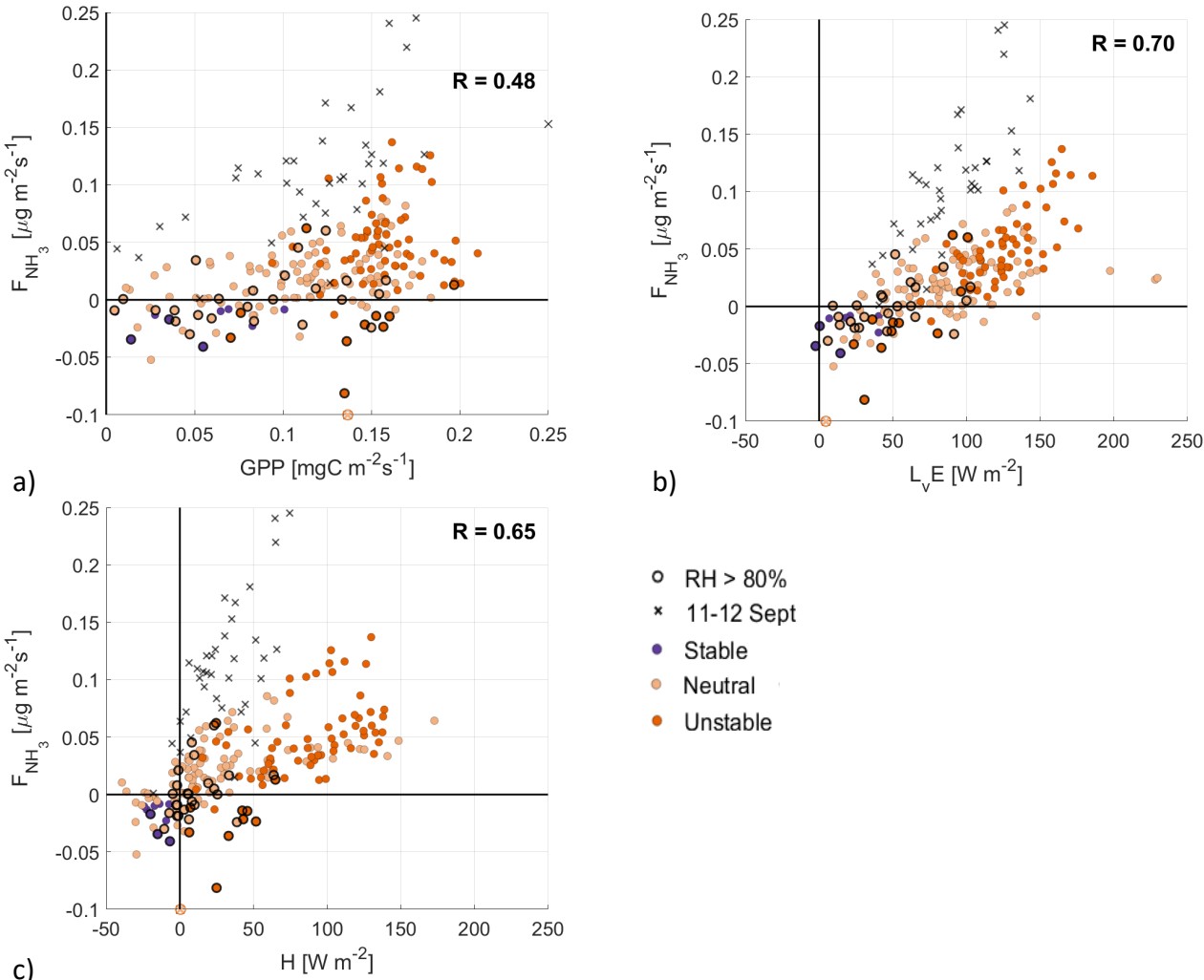

**Figure 4.** Scatter plots of $F_{NH_3}$ against the GPP (a), $L_vE$ (b) and H (c), with the colors indicating the atmospheric boundary layer (ABL) stability. Highlighted by black circles are observations with a RH >80 %, where deposition through the external leaf path can still play an important role. The black crosses are observations from the fertilization event observed on 11 - 12 September.

a role as well. Assuming a vegetation cover of 90% for grass, soil evaporation contributes with estimations that range from 10 to 30%. Despite this, the use of net $L_vE$ is acceptable as an indicator for the transpiration process. Note further that the observations with high relative humidity generally correspond to low $L_vE$ and that again unstable conditions correspond with high $L_vE$ values, related to the VPD between the leaf and stomata, and the atmosphere.

When plotting $F_{NH_3}$ against H, two branches are found in the spread of the data, with a third branch being formed by the filtered out fertilization event on 11 - 12 September (black crosses). The smaller branch, with $F_{NH_3} > 0.1$ $\mu g$ m$^{-2}$s$^{-1}$, could point towards another (weaker) fertilization event. Still, the second highest positive correlation is found at 0.65, indicating that the natural diurnal variability indeed plays an important role. Note that most of the measurements with high relative humidity are clustered around H = 0 W m$^{-2}$, i.e. there is little transfer of heat between the surface and atmosphere.

Based on the three scatter plots we find the highest correlation between $F_{NH_3}$ and $L_vE$. Together with the diurnal variability of $F_{NH_3}$, transitioning from nighttime deposition to daytime emission from 8:30 to 16:30 UTC, this is the second indication towards stomatal emission of NH₃, opposed to emission from fertilization or animal droppings. However, the moderate correlation between $F_{NH_3}$ and H indicates that the diurnal variability of the fluxes influences the correlation coefficient. Finally, we want to mention the observations on 11 - 12 September, which support the interpretation of the scatter plots in showing how fertilization events affect our analysis.

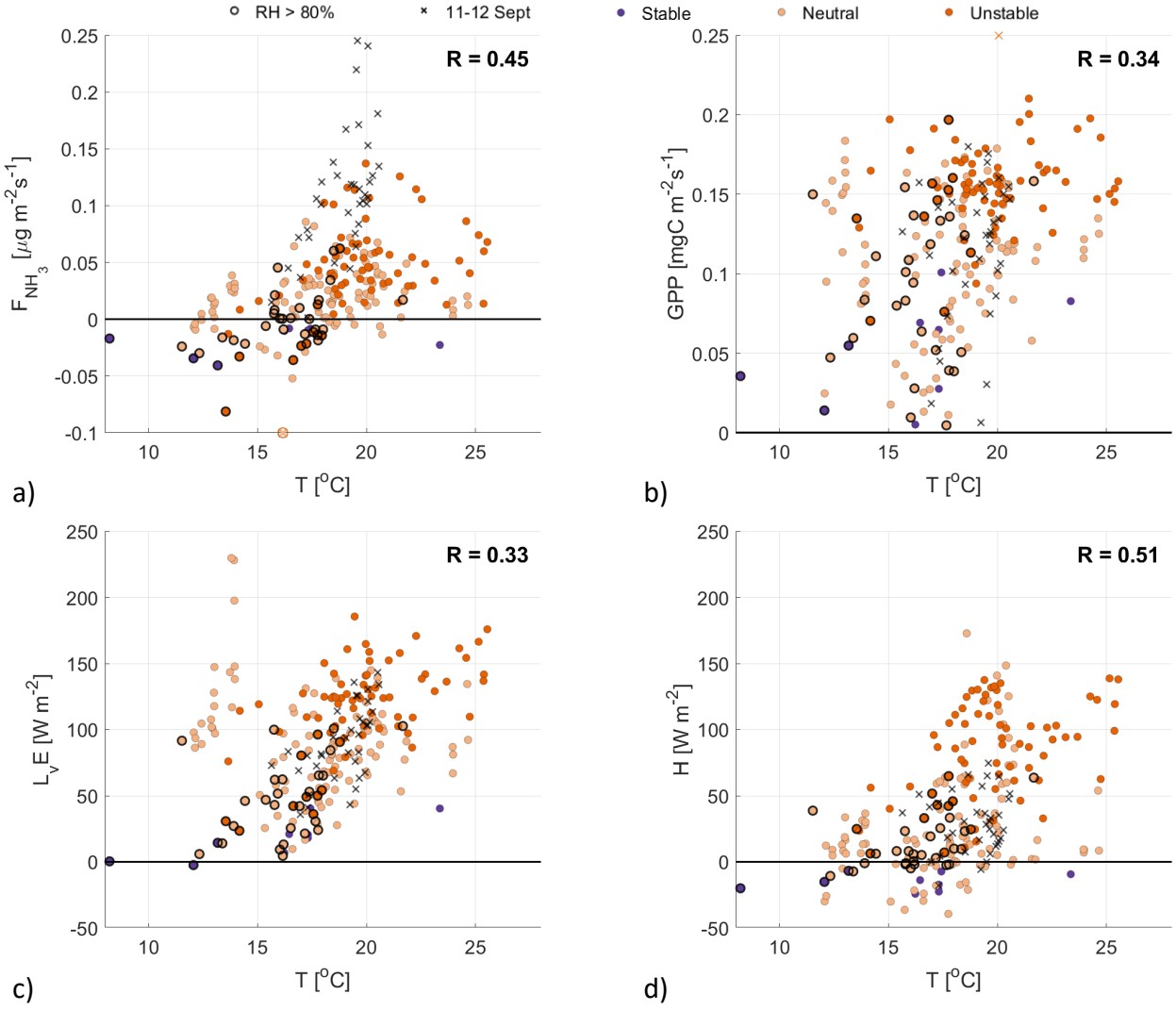

**Figure 5.** Scatter plots of the temperature against $F_{NH_3}$ (a), GPP (b), $L_vE$ (c) and H (d), with the colors indicating the ABL stability (see Figure 4 for legend). Highlighted by black circles are observations with a RH > 80 %. The black crosses are observations from the fertilization event on 11 - 12 September.

## 3.2 The dynamic vegetation response to varying meteorological conditions

### 3.2.1 The dynamic response to temperature

We further investigate the stomatal exchange of NH₃ by analyzing the response of $F_{NH_3}$ to varying meteorological conditions. The optimal conditions (PAR, T, VPD) for photosynthesis are different for different vegetation types (Gates, 1980; Jacobs, 1994; Vilà-Guerau de Arellano et al., 2015). Starting with the 2.8 meter temperature (T) in Fig. 5, we find a large spread for all four surface fluxes, resulting in low positive correlations (0.33 - 0.51). The lowest correlation coefficients are found for GPP and $L_vE$, indicating that temperature has little impact on the opening and closing of the stomata. Slightly higher correlation is found for $F_{NH_3}$, which we attribute to the relation between the stomatal compensation point and the NH₃ flux, discussed in Section 2.5. Note that the NH₃ emissions on 11 - 12 September stand out as outliers in Fig. 5a, while being average for the other three subplots.

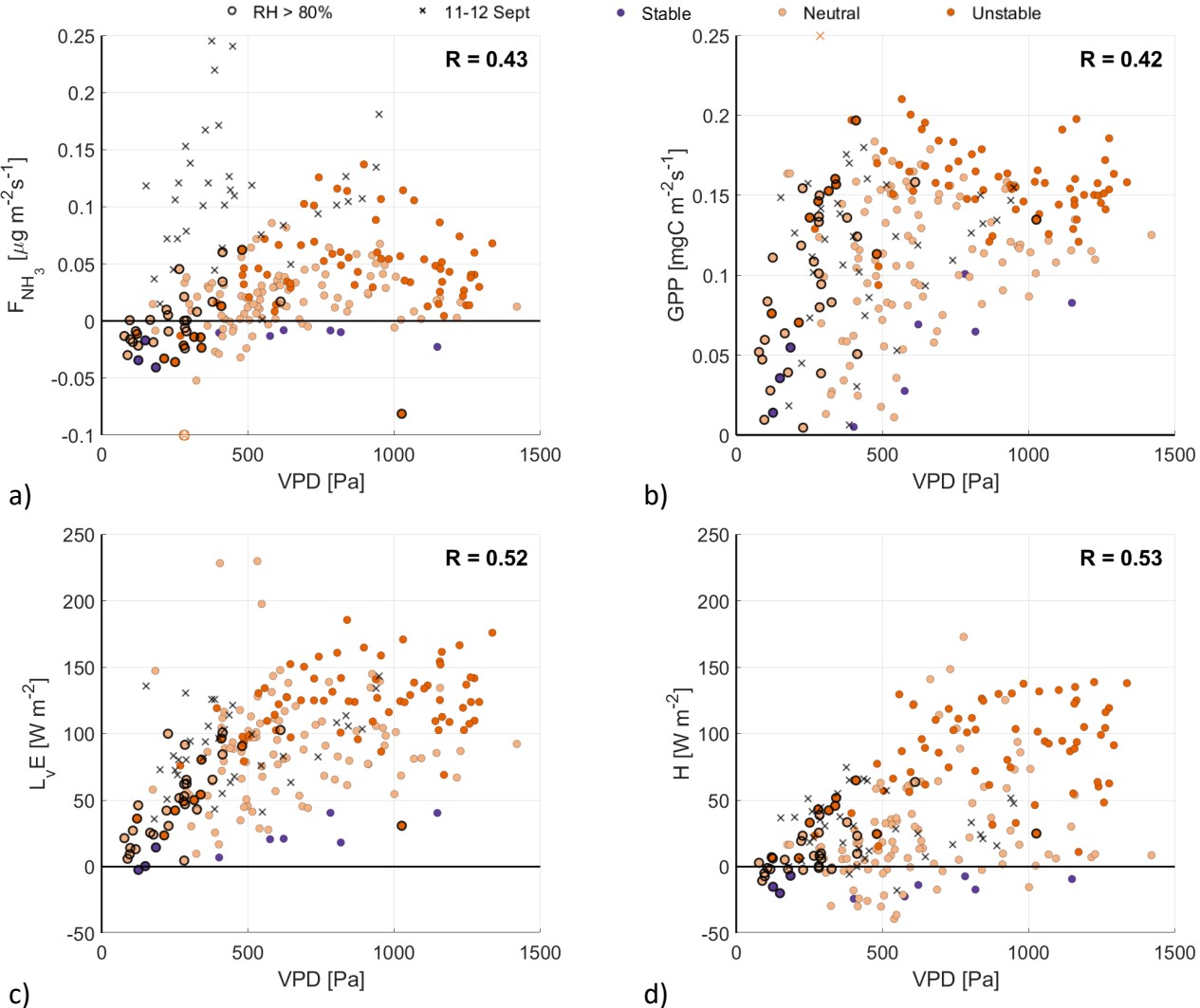

**Figure 6.** Scatter plots of the VPD against $F_{NH_3}$ (a), GPP (b), $L_vE$ (c) and H (d), with the colors indicating the ABL stability (see Figure 4 for legend). Highlighted by black circles are observations with a RH > 80 %. The black crosses are observations from the fertilization event on 11 - 12 September.

### 3.2.2 The dynamic response to VPD

Moving on to analyzing the response of the four fluxes to the VPD, we find moderate correlation coefficients (0.42 - 0.53) in Fig. 6. $L_vE$ shows in Fig. 6c a non-linear relationship with the VPD, called the evaporation hysteresis (Zhang et al., 2014; de Groot et al., 2019). This hysteresis is driven by both the vegetation regulating the loss of water through evaporation, described in Section 2.2, and the time difference when the maximum values for $L_vE$ (12:00 UTC) and VPD (15:00 UTC) are reached. The same holds true for the other three fluxes ($F_{NH_3}$, GPP and H), as all three reach their maximum around noon.

Note that the observations of 11 - 12 September again are clear outliers in Fig. 6a, forming two branches in the scatter plot. Also standing out are several observations with $F_{NH_3} > 0.1$ $\mu$g m⁻²s⁻¹. These are the observations that appear as the small upper branch in the H - $F_{NH_3}$ scatter plot in Fig. 4c and, again, form their separate branch here in Fig. 6a. This further indicates that there is a second (weak) fertilization event in the filtered dataset of the RITA-2021 campaign.

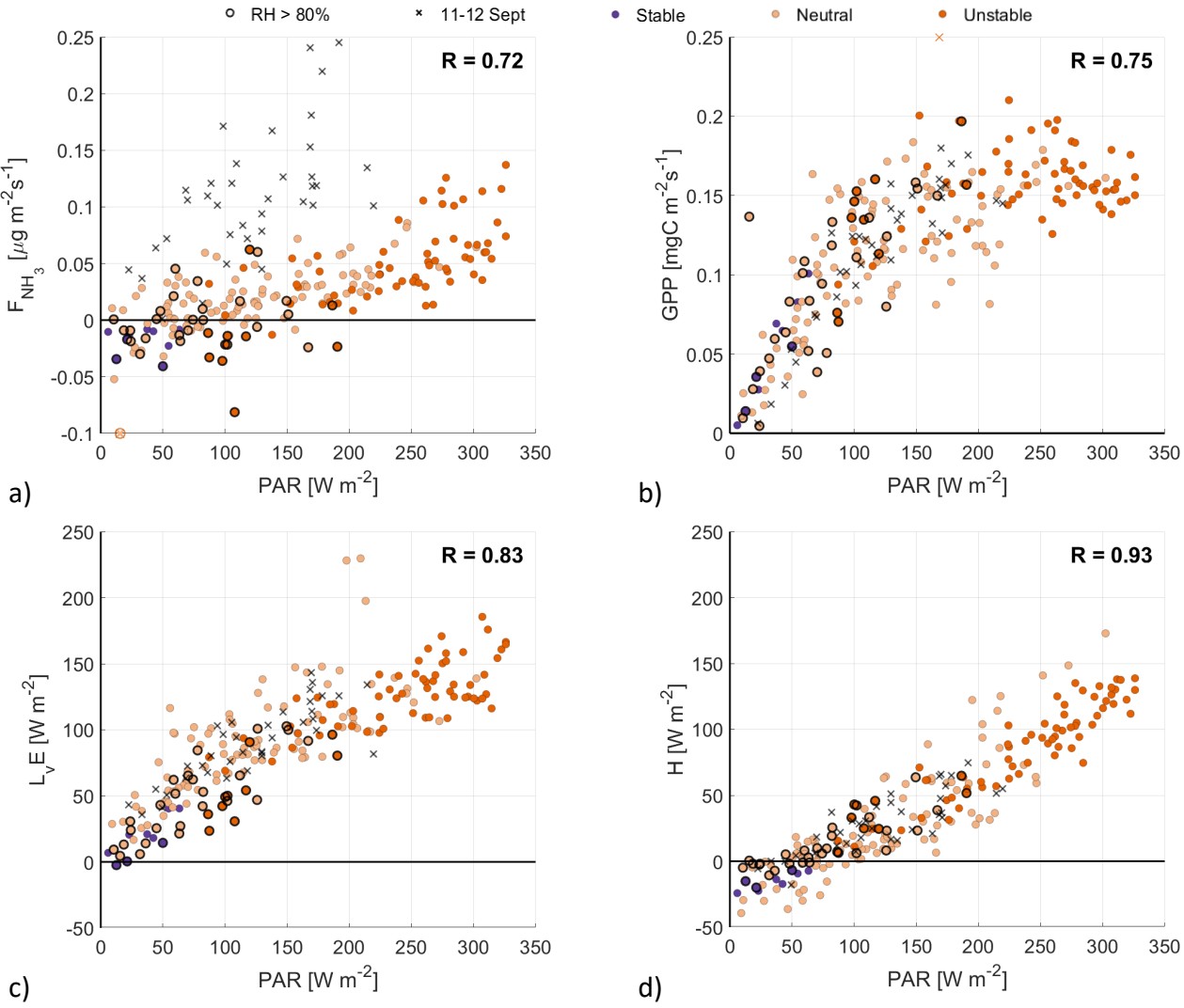

**Figure 7.** Scatter plots of PAR against $F_{NH_3}$ (a), GPP (b), $L_vE$ (c) and H (d), with the colors indicating the ABL stability (see Figure 4 for legend). Highlighted by black circles are observations with a RH > 80 %. The black crosses are observations from the fertilization event on 11 - 12 September.

### 3.2.3 The dynamic response to PAR

When relating the fluxes to PAR, we find high positive correlation coefficients for all four surface fluxes (0.72 - 0.93)(Fig. 7) indicating that PAR is the main driver of the dynamic vegetation response.

The GPP has a strongly non-linear response to PAR, as the GPP appears to reach a plateau for PAR > 150 W m$^{-2}$. There are several reasons for this GPP maximum. At constant temperature and PAR, the stomatal uptake of $CO_2$ will increase the concentration within the plant to the point that the $CO_2$ supply is no longer the limiting factor. The GPP then reaches a plateau of maximum photosynthesis rate (see Figure 6.13a Moene and Van Dam (2014)), similar to the observations in Fig. 7b. Additionally, the photosynthesis system can become light saturated for high PAR values, at constant temperature. Following this latter process, the GPP is expected to level off more gradually, compared to the plateau that is reached by $CO_2$ saturation (see Figure 6.13b Moene and Van Dam (2014)). Finally, the (partly) closing of the stomata in response to high VPD could also reduce the GPP. However, as the VPD typically reaches its maximum around 15:00 UTC (not shown), it is unlikely that this is a limiting factor for GPP at high PAR values, which peaks around noon. All these processes depend on the temperature, VPD and PAR, and can explain the vertical spread in Fig. 7b.

When taking a close look at the response of $L_vE$ to PAR, it is possible to distinguish two phases in Fig. 7c. First, for PAR values up to about 100 W m$^{-2}$, $L_vE$ increases linearly to roughly 75 W m$^{-2}$, related to the opening/closing of the stomata around sunrise/sunset. The second phase shows a more gradual linear increase of $L_vE$ to PAR. From the linear response and the small spread in Fig. 7c, we conclude that opening and closing of the stomata during the RITA-2021 campaign is governed by PAR and that the role of the VPD or temperature is small.

Similar to $L_vE$, the $NH_3$ flux generally shows a linear response; transitioning from weak deposition to emission as the stomata open in response to increasing PAR. The spread in the $F_{NH_3}$ response is larger compared to the $L_vE$ response, which results in the lowest correlation coefficient at 0.72. We attribute this spread to three factors: the relation between temperature and the stomatal compensation point, the variations in the $NH_3$ concentration and the measurements where RH > 80 % (black circles). Furthermore, observations where $F_{NH_3}$ > 0.1 $\mu$g m$^{-2}$s$^{-1}$, i.e. the possible (weak) fertilization event, again appear to form a second branch in the scatter plot. Based on the strong similarities between $F_{NH_3}$ and $L_vE$ in their response to PAR, we interpret the observed $NH_3$ emission as stomatal (re)emission from vegetation.

## 4 Discussion

Observations of the $NH_3$ flux after filtering, taken over 17 individual days during the RITA-2021 campaign, are characterized by day-time emissions. The measurement site at Cabauw is located on flat grassland in an agricultural area, with the nearby fields being actively managed and/or grazed upon. It is therefore possible that the observed $NH_3$ emissions originate from sources like fertilization events (e.g. manure application) or animal droppings. Clearly distinguishing between stomatal driven emission and volatilization of ammonia due to fertilization events is complex due to the contributions of different paths (soil versus plant) and non-linear effects (water vapor deficit dependence on temperature) that often offset each other. However, we identified $F_{NH_3}$ which are most likely due to a fertilization event and labeled these data as outliers, while keeping other doubtful points in the analysis. Next, we also marked $F_{NH_3}$ which could be due to exchange via the external pathway, once more trying to single out $F_{NH_3}$ due to stomatal exchange.

Indications towards stomatal emission is found in the diurnal variability of $F_{NH_3}$. The flux transitions from deposition to emission in the early morning around 8:00 UTC, reaches maximum emission around 12:00 UTC and transitions to deposition again just before sunset, around 16:30 UTC, as shown in Fig. 2c. Our interpretation of this diurnal cycle is the flux transitions from (nighttime) $NH_3$ deposition, through the external leaf path, towards emission through the stomatal path during the day. This diurnal variability of $F_{NH_3}$ shares similarities to the diurnal variability of the $CO_2$ flux. As the stomata open for photosynthesis in response to PAR, the $CO_2$ flux transitions from $CO_2$ respiration to stomatal uptake of $CO_2$. High correlation between $F_{NH_3}$ and $L_vE$ (0.7) and between $F_{NH_3}$ and PAR (0.72), further point towards stomatal $NH_3$ emission and a possible relation between $F_{NH_3}$ and the photosynthesis fluxes.

### 4.1 Critical analysis of RITA-2021 dataset

The conditions during the RITA-2021 campaign present a challenge for the analysis conducted in this study. The site is located in an active agricultural region, with several potential emission sources within only a few hundred meters to a couple of kilometers distance upwind of the measurement site. The fields next to the site are actively managed and nitrogen contents of the soil and vegetation can differ on a field-to-field basis. This high level of surface heterogeneity within the estimated footprint of the flux measurements (up to about 250 m, Table 2) adds an additional level of complexity to the analysis (Swart et al., 2023). Furthermore, there are several farms located within two kilometers of the site, some of which with yearly $NH_3$ emissions up

to 1200 kg year$^{-1}$. Studies on the blending distance (i.e. the distance at which a plume can be considered well-mixed with respect to the background) indicate that emission plumes from such strong local NH$_3$ sources can affect flux measurements over distances of a couple of kilometers (Schulte et al., 2022). In this study, at least one instance of strong local emissions has been identified as the fertilization event on 11 - 12 September. Other potential weaker events have been shown and discussed as well in Fig. 4a, 4c and 6a.

The analysis is further complicated by the complex meteorological conditions, characterized by frontal passages. As the miniDOAS setup was positioned anticipating winds from the south-west, the meteorology of the filtered data is characterized by frontal passages. As a result, most observations are taken under neutral stability conditions (60 %), with clouds and some rain showers. While rain events are filtered out, wet deposition by rain does lead to a sudden change in the NH$_3$ concentration and can lead to re-emission of NH$_3$ as the rainwater evaporates.

Finally, the south-western orientation of the instruments leads to a significant loss in the availability of data suitable for analysis. Historically, southwestern winds tend to be most common in September, but the wind direction during the campaign was highly variable. Filtering for unobstructed wind directions reduces the availability of viable data by 510 hours, i.e. 44 % of all measurement data. As a result, the observed range in the measurements presented in the Figures is strongly influenced by the natural diurnal variability of the variables. While we do address the role of the natural diurnal variability by including the sensible heat flux in our analysis, it does make the observed relations between F$_{NH_3}$ and the other variables somewhat speculative.

The high level of heterogeneity due to complex emission sources, the low data availability after filtering and the complex weather conditions make the RITA-2021 dataset unfavorable for establishing relationships between F$_{NH_3}$ and the CO$_2$ or water vapor flux. It also makes the dataset unsuitable to aid annual inventories. Yet, it highlights the importance of homogeneity of the NH$_3$ surface characteristics and that proximity of NH$_3$ emission sources should be considered as well when selecting a measurement site, in addition to the availability of high quality meteorological observations. Despite the challenges, the NH$_3$ measurements are of unprecedented high quality (Swart et al., 2023) and analyzing the unique dataset following our approach is still worthwhile; establishing relationships that significantly correlate with the main drivers of the stomatal aperture following current dynamic vegetation models.

## 4.2   Recommendations

Following the results presented in this study, we recommend a comprehensive approach to future NH$_3$ flux measurements, including observations of the CO$_2$ and water vapor flux as auxiliary measurements. The opening of the stomata for CO$_2$ uptake through photosynthesis allows for the exchange of several other gasses, including water vapor and ammonia. The process representations of photosynthesis have been widely researched and it has been better tested against sub-diurnal observations under different scales (Vilà-Guerau de Arellano et al., 2020) and such auxiliary observations can be used to further our understanding of NH$_3$ exchange through the individual exchange pathways, as was done for ozone deposition by Visser et al. (2021).

Furthermore, we recommend to analyze and compare observations of the NH$_3$ flux at different measurement (grassland) sites, similar to the intercomparison of CO$_2$ exchange measurements by Jacobs et al. (2007). For example, the F$_{NH_3}$ diurnal variability presented in this study significantly differs from measurements in 2013 at the Veenkampen meteorological site near the city of Wageningen (https://www.wur.nl/en/show/Weather-Station-De-Veenkampen.htm). Located only 50 km east, the diurnal variability of F$_{NH_3}$ at Veenkampen is characterized by weak morning deposition and strong afternoon deposition, up to about -0.3 $\mu$g m$^{-2}$s$^{-1}$, for clear sky conditions over unfertilized grassland (Schulte et al., 2021). At the Haarweg meteorological site, the predecessor to the Veenkampen, chemical wet denuder measurements of F$_{NH_3}$ in 2004 were characterized by strong deposition in the early morning, attributed to morning dew, and weak stomatal emissions in the afternoon (Wichink Kruit et al., 2007). The differences between observed diurnal variability in these three studies stress the high variability at the local and regional scales and highlight the need for long term high-resolution F$_{NH_3}$ observations at multiple locations.

Efforts to further our understanding of the NH$_3$ exchange and its diurnal variability are already being made. The miniDOAS setup used in the RITA-2021 will be taking long-term ($> 1$ year) observations of the NH$_3$ flux at the Veenkampen meteorological site, starting in the spring of 2023. This year-long record of high-resolution F$_{NH_3}$ observations will be analyzed, alongside a wide range of meteorological and turbulent measurements, including the CO$_2$ and water vapor flux, aiming to improve the parameterization of the NH$_3$ surface-atmosphere exchange. The collocation of surface and upper-atmospheric observations (Vilà-Guerau de Arellano et al., 2023) is key to obtain a comprehensive and complete understanding of NH$_3$ flux. The analysis can be taken one step further in the context of the Ruisdael Observatory project, following a process analysis combining the observations with both conceptual (Schulte et al., 2021) and high-resolution turbulent resolved models (Schulte et al., 2022).

## 5    Conclusions

We analyzed over a month of ammonia flux measurements (F$_{NH_3}$), taken during the RITA-2021 campaign at the Ruisdael Observatory at Cabauw. The analysis is centered around observations from the miniDOAS flux measurement setup, which applies the flux-gradient method to line average concentration measurements over a 22 m open-path at two heights. Our objective was to find relationships between the observed NH$_3$ flux and the main drivers of dynamic vegetation response, linking ammonia exchange through the main three variables that control the stomatal pathway to processes due to photosynthesis. The process of photosynthesis has been more widely studied and therefore establishing robust relationships between photosynthesis drivers closely linked to stomatal aperture and NH3 surface exchange enable us to determine and quantify the role of this path in emiting or depositing ammonia.

After filtering, the observed F$_{NH_3}$ is characterized by daytime emissions, averaging at about 0.05 $\mu$g m$^{-2}$s$^{-1}$, and nighttime deposition of about -0.05 $\mu$g m$^{-2}$s$^{-1}$. We compare the NH$_3$ flux to the from observations inferred CO$_2$ uptake by vegetation and the net observed exchange of water vapor, represented by the gross primary production (GPP) and net latent heat flux (L$_v$E), as well as the sensible heat flux (H) which is only indirectly related to the dynamic vegetation response. Here, we find high and significant correlation between the observed daytime NH$_3$ emissions and L$_v$E (0.70) and the photosynthetically active radiation (PAR, 0.72). These results provide a first-order quantification of how NH$_3$ exchange could follow similar paths as the exchange of CO$_2$ and H$_2$O through plant processes regulated by the stomatal aperture.. It shows that auxiliary and collocated flux measurements of CO$_2$ and water vapor are appropriate variables to distinguish stomatal NH$_3$ exchange from non-stomatal exchange.

The analysis presented in this study is hampered by the challenging conditions during the RITA-2021 campaign. However, despite these conditions, the comprehensive approach presented in this study paves the way for the potential of combining high-quality NH$_3$ observations with auxiliary flux measurements of CO$_2$, water vapor and other meteorological variables. By organizing and analyzing the observations guided and constrained by the main meteorological drivers controlling the assimilation and transpiration in grass fields, we managed to attribute the observed NH$_3$ emission to processes and variables associated to stomatal exchange and identify outliers. In order to establish more robust relations between NH$_3$ and the photosynthesis fluxes, the proposed framework in this study should be applied to measurements that are still representative of the nearby sources and sinks, but insuring a blending distance that guarantees that these singular sources and sinks contributions are properly mixed with the NH$_3$-background concentration. These distances range from 1000 $m$ to 3000 $m$ (Schulte et al., 2022). Further, longer time series are needed in order to make a more robust distinction between days with and without the influences of nearby sources. Our findings and framework over grasslands are a first step to confirm patterns and relationships between meteorological drivers and NH$_3$ exchange, but should be extended to longer and more dedicated field campaigns, including other ecosystems. The results presented in this study already indicate that there is room to find such patterns.

**Author Contributions**

RS: Conceptualization, Formal Analysis, Investigation, Methodology, Visualization, Writing - original draft preparation. JVGdA: Conceptualization, Writing - Review and Editing, Supervision. SJR: Methodology and Investigation (NH$_3$ flux data) SvdG Investigation (NH$_3$ flux data). JZ: Investigation (CO$_2$ flux and meteorological data). MCvZ: Conceptualization, Funding Acquisition, Writing - Review and Editing.

**Competing interests**

The authors declare that they have no conflict of interest.

## Appendix A:  An alternative way of calculating ecosystem respiration

In 2.2, we describe our approach to arrive at an estimate of GPP using observations only. Here, we examine the potential impact of using a regression model to describe the ecosystem respiration to examine the potential impact of using a different method on the results. We calculated GPP by describing ecosystem respiration as a function of air temperature using the exponential regression model of Lloyd and Taylor (1994), hereafter LT94:

$$ER = R_{10}exp(E_0(\frac{1}{T_{\text{ref}} - T_0} - \frac{1}{T - T_0}) \tag{A1}$$

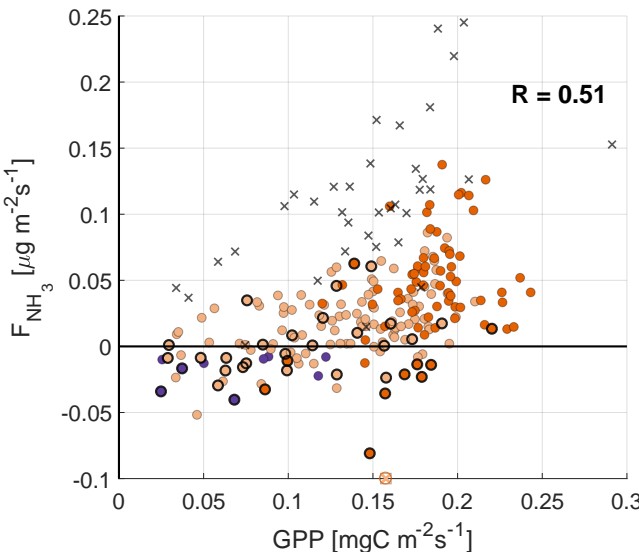

**Figure A1.** Scatter plots of $F_{NH_3}$ against the GPP, with the colors indicating the atmospheric boundary layer (ABL) stability. Highlighted by black circles are observations with a RH >80 %, where deposition through the external leaf path can still play an important role. The black crosses are observations from the fertilization event observed on 11 - 12 September.

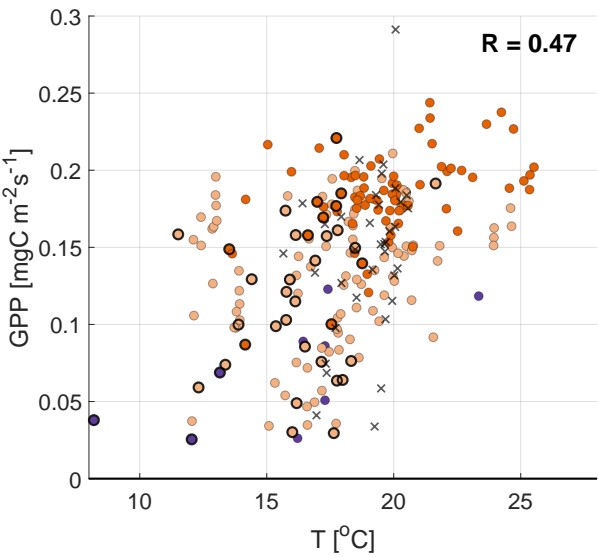

**Figure A2.** Scatter plots of the temperature GPP, with the colors indicating the ABL stability (see Figure 4 for legend). Highlighted by black circles are observations with a RH > 80 %. The black crosses are observations from the fertilization event on 11 - 12 September.

Where $R_{10}$ is the reference respiration at reference temperature $T_{ref}$ (set to 10 °C). To avoid over-paramaterisation, $T_0$ is set to -46.02 °C, as in LT94. $E_0$ is an empirical parameter related to the activation energy. Using the night-time data collected during the campaign, filtered for $u_* \geq 0.1 ms^{-1}$, and quality flag of 0 (Mauder and Foken, 2006), we obtained values of 5.3 for $R_{10}$, and 124 for $E_0$. In doing so, correlation coefficients for the corresponding panels in Figure 4 -3.2.3 of the main text
5 slightly improved. Figures A1 - A4 show the scatter plots, using this alternative formulation of GPP.

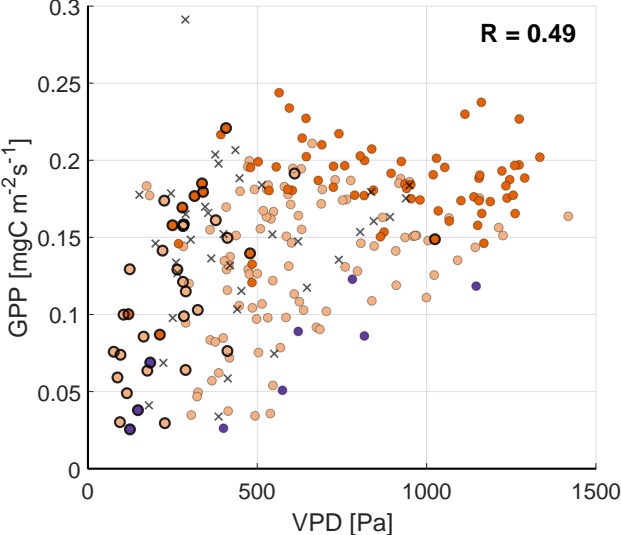

**Figure A3.** Scatter plots of the VPD against GPP, with the colors indicating the ABL stability (see Figure 4 for legend). Highlighted by black circles are observations with a RH > 80 %. The black crosses are observations from the fertilization event on 11 - 12 September.

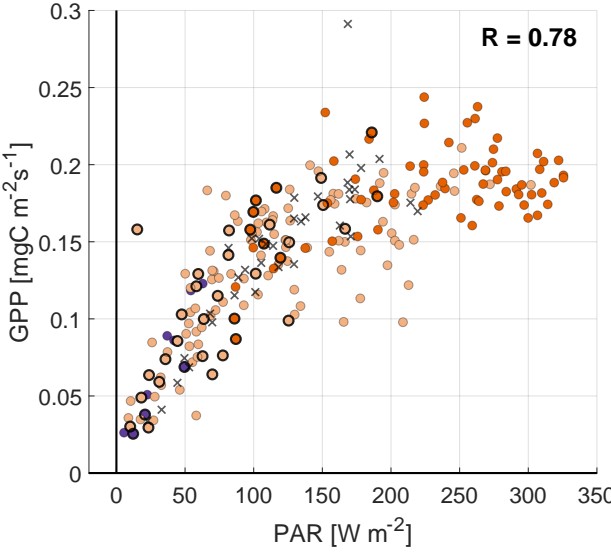

**Figure A4.** Scatter plots of PAR against GPP, with the colors indicating the ABL stability (see Figure 4 for legend). Highlighted by black circles are observations with a RH > 80 %. The black crosses are observations from the fertilization event on 11 - 12 September.

*Acknowledgements.* Ruben B. Schulte PhD project was supported by RIVM within the framework of project 36.7: Monitoring of dry ammonia deposition; financed by the Dutch Ministry of Agriculture, Nature and Food Quality. We thank the Royal Netherlands Meteorological Institute (KNMI) for site access and assistance during the RITA -2021 campaign. All our colleagues of RIVM (D. Swart, S. Berkhout, R. van der Hoff and M. Haaima) and of TNO (A. Hensen, P. Wintjen, A. Frumau and P. van den Bulk) involved in this campaign are gratefully acknowledged for their support, dedicated work to make the campaign a success and fruitful discussions during the analysis of the data. This article has been accomplished by using data generated in the Ruisdael Observatory, a scientific research infrastructure which is (partly) financed by the Dutch Research Council (NWO, grant number 184.034.015).

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
