# Peer review of "Observational relationships between ammonia, carbon dioxide and water vapor under a wide range of meteorological and turbulent conditions: RITA-2021 campaign"

_EGUsphere, 2023_

## Author Comment (AC1)

The manuscript by Schulte et al. presents campaign data of ammonia measurements by miniDOAS instrumentation in a gradient setup together with CO2 and water vapor fluxes and meteorological parameters from the Ruisdael Observatory at Cabauw in the Netherlands. The authors investigate relationships between fluxes and meteorology and try to establish a link between NH3 and CO2 exchange through the stomatal pathway.

The approach is good and the dataset useful. Most of the text is well written and easy to follow. Methods are robust, figures are clear and easy to grasp. Ammonia flux measurements are still highly experimental and it is good to see more campaign data being presented using relatively novel methods with high accuracy concentration readings. The shown filtering scheme for high quality data assurance applied in the main analysis is thoroughly done. However, there are some shortcomings that need to be fixed to make this a study of broader interest for the ammonia flux and modelling community.

**Major comments**

- The campaign was conducted in a region with agricultural activity. The authors state that the observed emissions are likely originating from fertilization or animal droppings. This means that much of the exchange is due to volatilization directly from the surface (soil or fertilizer). The analysis, however, is solely associated with stomatal exchange of ammonia. Both pathways are known to exhibit distinct diurnal patterns, basically following the course of temperature, radiation, and turbulence, making a differentiation between stomatal and non-stomatal exchange difficult. In this view, I suggest to revisit the aims of the paper. A number of correlations between fluxes and meteorological parameters are shown, which is surely useful and some findings are impressive. But I clearly miss a coherent storyline apart from "establish relationships". What is the focus of investigation? Much can be fixed by rephrasing, refining conclusions, and interpretation of the validity of the findings.

- The focus of the investigation is to study the relationships between photosynthesis and the NH3 exchange through the stomata. To this end, we ask the following question: does stomatal NH3 exchange response similarly to the environmental drivers of dynamic vegetation response as GPP and LvE? Hereby GPP is representing CO2 uptake and LvE evaporation which is subsequently taken as a proxy for the plant transpiration process of vegetation due to grassfield covers 90% of the surface. Our study is guided by the analysis of measurement data itself, and keeps the use of models and thereby assumptions to interpret the data to a minimum. We realise that our data is limited due to the weather conditions and the complexity associate to

the nearby multiple sources of ammonia. However, the data set is comprehensive in terms of meteorology, carbon dioxide and ammonia flux measurements. We have also been very rigorous and transparent in our data criteria and analysis. In that respect, the analysis and the proposed method acts as a proof of concept which can serve as an example and pave the way to future campaigns. We will make this more clear in the introduction by adding the following text at the end:

'As our data set is limited due to the diversity of weather conditions and the complexity associated to nearby multiple sources of ammonia, our analysis acts as a proof of concept. Serving as an example for the need of combined high quality NH3 flux measurements with auxiliary measurements of CO2, water vapor fluxes and other meteorological variables. As such we decided to guide our analysis on observations and keep the use of representation of processes to interpret our data to a minimum. Thus, CO2 uptake is estimated by subtracting the average campaign night time CO2 flux from the observed CO2 flux instead of using a representation of the CO2 soil efflux. Further, we use LvE measurements representing net evaporation; by neglecting the soil evaporation part we use LvE as an indicator for the transpiration process. '

There is a misunderstanding that we state that the observed emissions are likely originating from fertilization or animal droppings. Instead we are *aware* that the observations could be influenced by nearby emission sources caused by sporadic fertilization events. In that respect, our filtering approach aims and largely succeeds to exclude these situations from our analysis. The same applies to discriminating between the stomatal pathway and other pathways of ammonia exchange: we have entangled these by marking observations in the figures where ammonia exchange via the external leaf pathway could play a role. Thirdly, in order to complete the analysis we have included the sensible heat H since it is the main drivers of turbulent convection during the day, and plays a key role in the local mixing and turbulent transport. H influences the dynamic vegetation response through the temperature gradient above the grass. If NH3 flux is regulated by opening and closing of stomata, this shows in a different comparison between NH3 flux and GPP/VPD compared to H which mainly depends on the temperature gradient (see figure 5).

- There seems to be a mixture of the terms "evaporation", "transpiration", "evapotranspiration", and "latent heat flux". Please check throughout the manuscript. It is really confusing. See also specific comments.

  The word evapotranspiration was used twice in the article, which we changed to evaporation instead. We explained more clearly in the text that we use LvE for the net evaporation that integrates plant transpiration and soil

evaporation. Hereby we follow the ideas of Miralles et al., 2020 On the Use of the Term "Evapotranspiration" Water Resources Research, 56, e2020WR028055. https://doi.org/10.1029/2020WR028055

- Ecosystem respiration strongly depends on soil temperature. Taking a campaign average to derive gross primary productivity may induce considerable uncertainty. Why didn't you use one of the well-established partitioning methods of the flux community? See specific comments for further details.

  We explained our reason for taking a campaign average to derive GPP above. Following the comment by the reviewer, in order to study how applying a partitioning method would change our analysis we calculated GPP by describing ecosystem respiration using the exponential regression model of Lloyd and Taylor (1994) with values for R10 and Ea fitted to the data collected during the campaign. In doing so, correlation coefficients for the corresponding panels in Figure 4 -7 slightly improved. However in our opinion, this doesn't justify steering away from the original idea of the article of using observations only. We will, however, include the material presented below in a supplement to the manuscript and discuss it  shortly in the main text.

|  | R value original | R value |
|---|---|---|
| Figure 4a | 0.48 | 0.51 |
| Figure 5b | 0.34 | 0.47 |
| Figure 6b | 0.42 | 0.49 |
| Figure 7b | 0.75 | 0.78 |

[Figure]

[Figure]

Figure 4a and 5b with the ecosystem respiration calculated using the exponential regression model of Lloyd and Taylor (1994)

[Figure]

Figure 6b and 7b with the ecosystem respiration calculated using the exponential regression model of Lloyd and Taylor (1994)

**Specific comments**

- Page 1, Line 14: "flux representations", what is meant here? Flux representations in models? Please clarify. 'in models' added

- Page 2, Line 49: How was the accuracy of the 30-min average concentration determined? Is it a statistical parameter based on a calibration procedure? Is the number coming from own test? Please provide more information.

In the manuscript, we state "The 30 minute average NH3 concentrations have an accuracy of 0.01 µg m-3". This statement has been altered based on what has been reported in Swart et al. (2023). New text reads: "The 30 minute average NH3 concentrations have an accuracy of 3% (e.g. 0.15 µg m-3 at the median $NH_3$ concentration of 5 µg m-3 during the campaign, for further details see Swart et al., 2023.".

- Overall, Figure 1 is very informative, but what is meant by "larger structures" (see caption)? Text changed to explain that larger structures denote the tower and containers. Also, in the legend, is the unit kg N per year or kg NH3 per year? Text changed to explain that the unit is kg NH3 per year.

- Page 3, Line 8: What has the McDermitt et al. (2011) paper to do with standard Fluxnet methodology? The paper is about a novel open-path methane instrument. Please check.

The sentence has been changed to "The flux calculation procedure followed the general best practices as applied across the FluxNet network (e.g. Mauder et al,

2021) including co-ordinate rotation Wilczak et al. (2001), spectral corrections for both filtering (Moncrieff et al., 2004) and low pass filtering (Moncrieff et al. 1997) and addition of the Webb–Pearman–Leuning density term (Webb et al., 1980).'

- Table 1 is very nice, but it should be described that the given filters are applied in series and after applying all of them, 9% of the data pass the filter (if I got it correct). Otherwise, it could lead to misunderstanding. Done

- Page 4, Line 11: How do you get to a number of 0.01 ug m-2 s-1? Please describe the procedure. And what exactly do you mean by "accuracy"? Is it the flux detection limit?

  In the manuscript, we state "With these three filters, we ensure the quality of the ammonia measurements, observing the NH3 flux with an accuracy of 0.01 µg m-2s-1.". This statement was not substantiated. Text will be changed to: 'With these three filters, we ensure the quality of the ammonia measurements, observing the NH3 flux with an average precision of 0.015 µg $NH_3$ m-2s-1 (1σ; for further details see Swart et al. (2023)).'.

  In Section 2.1, a little more information has been added about the performance of the miniDOAS instruments. After "The flux measurement setup uses two miniDOAS instruments, which measure the concentration over parallel paths at different heights, i.e. 0.76 m and 2.29 m respectively." the following sentence has been added:
  "Regular intercalibrations between the miniDOAS instruments allowed quantification of and correction for any potential bias between the two instruments. The remaining random uncertainty in delta $NH_3$ was 0.088 µg $NH_3$ m-3 (1σ; for further details see Swart et al. (2023)).".

- Page 4, Line 18: Sentence starting with "While these processes…": What's the message? Either elaborate a bit more and add context or delete. Sentence changed to: These processes occur at the leaf scale (micrometer or millimeter level) and as such require a representation of photosynthesis and stomatal aperture that requires to be evaluated with observations Vila et al., 2020. The upscaling to the canopy level, allows it to be compared with observations inferred from eddy-covariance such as GPP (Filter 4).

- Page 4, Line 33: NEE = GPP + RESP, check sign convention. GPP and RESP are usually given as positive fluxes, then it should either be NEE = GPP – RESP (biological sign convention) indicating that a positive NEE represents a net carbon gain for the ecosystem or NEE = RESP – GPP (atmospheric sign convention) indicating that a positive NEE represents a net carbon gain for the atmosphere. Text changed, formula is given according to ecological sign convention.

- Page 4, Line 34: Average campaign nighttime flux? Why do you do that? Ecosystem respiration strongly depends on soil temperature. This may induce considerable uncertainty on your GPP estimates. Why didn't you use the one of the well-established methods in the flux community based on either daytime (Lasslop et al., 2010) or nighttime data (Reichstein et al., 2005) for partitioning measured NEE into GPP and RESP? See also Wutzler et al. (2018) and Pastorello et al. (2020). See the text and figures above in the reply to the third major comment.

- Page 4, Line 37: In the context of CO2, I think it is not "deposition". Please replace with "uptake" or "sequestration". Done

- Table 2 caption: "Any" number? Consider rephrasing to "at which  observational data passes the filters". Done

- Table 2 caption: I do not understand the whole sentence starting with "A requirement of non-zero...". Please rephrase.
  Sentence is changed to make clear that for GPP and flux footprint length only daytime data are used, i.e. night time data is excluded. Also 'Daily maximum GPP' has been corrected to 'daytime maximum GPP'.

  Table 2 caption: What is a "footprint anemometer"? We removed 'anemometer' and added the following explanation: in the table itself: Daily maximum flux footprint length (70%) In the caption: the variable "Daily maximum flux footprint length (70%)" refers to maximum upwind distance in meters encompassing the source area that contributed 70% of the measured flux.

- Table 2: The entry "Daytime maximum sonic anemometer footprint (70%)" requires more explanation. See above

- Page 5, Line 7: Please describe what "are actively managed for the agricultural activity" means. Sentence changed to "Additionally the ground and surface water levels are actively managed in order to sustain optimal conditions for the agricultural activity in the area.'

- Page 5, Line 16: See my comment on Table 2. This sentence is hard to understand. I would at least suggest to add that 70% represents the value of the isoline confining the area that contributed 70% to the measured flux (if that is what you mean).
  Text replaced with: .'Additionally, Table 2 includes an estimate of the maximum daytime footprint determined using the sonic anemometer fluxes at a height of 2.8 m, following the method from Kljun et al. (2015). This footprint refers to the maximum upwind distance in meters encompassing the source area that contributed 70% of the measured flux and serves as a first-order approximation of the footprint of the NH3 flux measurements

- Page 5, Line 24: For clarification, I suggest to replace "observations" in the title by "concentrations". Word kept unchanged, the paragraph also shortly describes the NH3 gradient and NH3 fluxes, however the word 'General' is added in order to distinguish from the next paragraph

- Page 5, Lines 26-27: I think I understand what you mean with "long tail", but I'm not sure this is a good expression. Consider rephrasing to something like "The histogram shows a highly skewed distribution with most concentrations being lower than 7 ug m-3 and a strong frequency decline for values >7 ug m-3." Text changed to: The histogram is highly skewed and shows that most observed NH3 concentrations are below 7 µg m-3, however higher concentrations with a maximum value of 24.7 µg m-3 are also present.

- Page 6, Line 3: "the diurnal variability", do you mean "their diurnal variability"? Otherwise, where does "variability" refer to? Sentence changed to make this clear: As FNH3 is directly inferred from ΔNH3, the diurnal variability in Fig. 2c and d is very similar.

- Page 6, Lines 10-19: This is text book or literature repetition. Is it really needed? Together with the rest of Section 2.4 [we assume the reviewer means 2.5] it appears a bit incoherent and as a sequence of rather loose facts. Lines 10-19 have been deleted. Some of the information has been added to the lines below, but only those directly relevant to the text.

- Page 6, Line 20: Why sonic temperature? It is not the same as air temperature. During the time of the analysis air temperature data was unavailable so we had to use sonic data instead. However, we used the corrected air temperature as calculated by the EddyPro software. Text in the manuscript will be addressed to clarify this.

- Page 6, Line 36: Sentence starting with "While there were only small variations...": check grammar, there is something wrong. 'Are' added to the sentence.

- Page 6, Line 40: "observations" – do you mean "concentrations"? Yes, observations replaced by concentrations.

- Page 6, Lines 47-50: I'm not sure I can follow the reasoning here. LvE is not just transpiration, but also evaporation from soil and water droplets, which has nothing to do with stomatal exchange of water. I think the two sentences are not wrong as they are, but I don't understand the message.
As explained above more clearly than in the original manuscript we use LvE as a proxy for transpiration since roughly 70% of LvE represents transpiration.

- Figure 2 caption: Add at the end: "...i.e., negative numbers indicate deposition and positive numbers indicate emission." Done

- Figure 3 caption: "observed NH3" – "observed NH3 concentration"? Changed

- Page 8, Line 16: "evaporation"? Aren't you talking about "transpiration"? Plotted in Figure 4b is LvE measured by eddy covariance, which is evapotranspiration, i.e., evaporation plus transpiration, right? Please clarify. Evaporation changed into transpiration. In the second sentence hereafter we note that LvE actually represents net LvE, i.e. net evaporation since soil evaporation plays a role as well. To make it more clear that LvE is used as a proxy (albeit not a perfect one) for the transpiration process we added the following sentence: Despite this, the use of LvE is acceptable as an indicator for the transpiration process.

- Page 8, Line 20: "between 10 and 30%" – where do these numbers come from?

  The numbers come from a study with the CLASS model in which we have evaluated satisfactorily the net evaporation against observations. Cabauw land use is covered for 90% with grass. With this values, we obtain estimations of 10-30% on soil evaporation and the rest of net LE corresponds to the assimilation of CO2 by grass. See also figure 2 in J. Vila et al., 2012 *Modelled suppression of boundary-layer clouds by plants in a CO2-rich atmosphere.* Nature Geosciences DOI: 10.1038/NGEO1554

- Page 9, Line 1: "...the diurnal variability influences the correlation coefficient" – What's the reasoning here? Please explain. Have you checked for hysteresis? Looking at Figure 4c, it may well be that on single days FNH3 is lagging behind H by probably a few half hours causing a hysteretic relationship between the two. Accounting for the lag (if there is one), may significantly increase the correlation coefficient.
  We are aware that hysteresis can play a role in our analysis as discussed in section 3.2.2. The patterns of these hysteresis depend also on external processes like the presence of clouds and advection, which is out of the scope of the study. For this case we think that the FNH3 points around 0.1 and above are an indication of a weaker fertilization event. However, we found the evidence not strong enough to mark them as outliers as we did with the fertilization event of 11-12 sept.

- Figure 5: What do correlations between temperature and LvE, GPP, and H tell us about drivers of and correlations between FNH3 and GPP, ET?
  We consider temperature, PAR and VPD to be the drivers of the dynamic vegetation response and a such compare FNH3, GPP and LvE with these three drivers. Our analysis shows that correlations are highest for PAR and lowest for temperature.

- Figure 6 caption: Has "ABL" been defined before? Definition of ABL has been added to the caption of figure 4.

- Page 11, Line 3 and Line 4: "evaporation" – again, aren't we looking at evapotranspiration? Please make it clear here and throughout the manuscript that there is a difference between transpiration and evaporation,

and that LvE represents the sum called evapotranspiration.
See our earlier explanation of our use of the term evaporation.

- Page 13, Lines 2-6: Needed? This is well known and should be shortened. Text has been shortened.

- Page 13, Lines 30-43: See my major comment on this topic.
See our response to major comment 1. We will rewrite these lines and explain that we are able to link the observed ammonia emission to temperature, VPD and PAR which can be a first step to improve parametrizations of FNH3 as a function of these drivers and connect it to well established representations of CO2 uptake.

- Page 14, Lines 18-25: I appreciate the honesty here and that the limitations of the study are clearly pointed out. However, I disagree with the statement that the ammonia flux could be linked to stomatal exchange. Page 15, Lines 26-27: "we managed to attribute the observed NH3 emission to stomatal exchange and identify outliers" – again, I'm not convinced.

  We will address this point and rewrite the specific lines of the manuscript with less strong claims (see also comment above).

- Conclusion section: Most of the section sounds like a summary with many repetitions that the reader has seen before. Please shorten and concentrate on real conclusions, i.e., what have we learned, what is new?

  We will follow the advice of the reviewer and shorten the conclusions.

**Technical corrections:**

- Page 1, Line 14: Remove the second "the". Done

- Page 1, Line 14: "dependencies" Done

- Page 1, Line 20: Remove "to" before "evapotranspiration". Done

- Page 2, Line 27: Either "These surface exchanges" or "This surface exchange". Done

- Page 4, Line 4: "of" between "reduction" and "data" missing. Done

- Page 4, Line 7: "by" between "characterized" and "frontal" missing. Done

- Page 4, Line 15: Singular "pathway". Done

- Page 4, Line 16: Check phrasing. What depends on RH? The external leaf pathway? If so, consider replacing "depending" with "and depends on". Done

- Page 4, Line 19: "millimeter"? Done

References:

Lasslop et al. (2010), https://doi.org/10.1111/j.1365-2486.2009.02041.x

Pastorello et al. (2020), https://www.nature.com/articles/s41597-020-0534-3;

Reichstein et al. (2005), https://doi.org/10.1111/j.1365-2486.2005.001002.x

Wutzler et al. (2018), https://doi.org/10.5194/bg-15-5015-2018

**Citation**: https://doi.org/10.5194/egusphere-2023-1526-RC2

---

## Author Comment (AC2)

**Reviewer 1**

**general comments**

This paper shows the relationship between stomatal activity and dynamics with respect to ammonia, exploring the correlations between several environmental variables involved in the process. I found the topic very interesting, and the work is well written: the data quality is very good in my opinion, and the data are explored very carefully, and discussed thoroughly. The authors addressed with the appropriate references the arguments posed by the study.

The weak point of this study is the poor temporal representativity of the dataset: this aspect is exposed in section 4.1, which I completely agree with. I think despite of the impossibility of setting new relationships to model the NH3 dynamics, this dataset provides a very useful verification of the known environmental dynamics, and especially shows very clearly what are the issues with NH3 measurements, setting a good standard for potential new measurements of NH3 fluxes.

Given the good quality of the paper, I agree with the authors that it is a shame the amount of data that needed to be rejected is so large: however, I believe this work is very worth publishing, not only for the scarcity of datasets on atmospheric ammonia fluxes in general (and even less of these standards!), but also for the analysis structure of the data, that provides a good methodology to be used not only by the future measurements that are taking places at the same site. Therefore I recommend its publication almost in its current shape, and I list below some minor points.

**specific comments**

The authors operate a filtering procedure that leads to considering 9% of all data valid for the evaluation of ammonia dynamics. While I understand the logic of excluding data for all the reasons listed, I think eliminating 91% of data is a fierce manipulation exercise. The conclusions of such reasoning are, by definition, not representative of the behaviour of the vegetation as such, but of particular conditions. This is addressed in the discussion and conclusions, but I suggest to reinforce the fact that this kind of dataset should not be used to aid annual inventories, perhaps in the abstract (but I'd leave it to the authors' and editor's choice where to insert it).

The following sentence: 'It also makes the dataset unsuitable to aid annual inventories.' Is added at pg 14, line 20.

 **technical corrections**

P4 L30: remove either "ammonia" or NH3 in the sentence.
Done

**Citation**: https://doi.org/10.5194/egusphere-2023-1526-RC1